# Application of Color Doppler with 3- and 4-Dimensional Ultrasonography in the Prenatal Evaluation of Fetal Extracardiac and Placental Abnormalities

**DOI:** 10.3390/healthcare11040488

**Published:** 2023-02-08

**Authors:** Kwok-Yin Leung

**Affiliations:** Obstetrics and Gynaecology, Gleneagles Hong Kong, Hong Kong, China; ky@kyleung.org

**Keywords:** Obstetrics, ultrasound, 3-D, 4-D, Doppler, STIC, placenta, twin

## Abstract

Using color Doppler flow imaging or high-definition flow imaging with three-dimensional volume or spatio-temporal image correlation (STIC) in the glass-body mode allows displaying both gray-scale and color information of the heart cycle-related flow events and vessel spatial relationship. Conventionally, STIC in the glass-body mode has been used to examine the fetal heart and assess heart defects. Recently, a novel application of STIC in the visualization of abdominal precordial veins and intraplacental vascularization in singleton pregnancies has been reported. The aim of this present review is to discuss the use of color Doppler with three- and four-dimensional ultrasonography in the evaluation of extracardiac, placental, umbilical cord and twin abnormalities with examples. The glass-body mode is complementary to conventional 2D ultrasonography. Further studies are required to investigate use of the glass-body mode in the assessment of intraplacental vascularization in singleton and twin pregnancies.

## 1. Introduction

It is a common practice to add color flow imaging (CFI) to gray-scale ultrasonography in obstetric scans to detect abnormal blood flow in fetal and placental abnormalities [1,2,3,4,5,6,7]. Various CFI modes were discussed in a recent review by Leung KY [8]. The characteristics of high-definition flow imaging (HDFI) and microvascular flow imaging (MVFI) are different from conventional modes, namely, color doppler flow imaging (CDFI) and power Doppler imaging (DPI) (Table 1) [8]. It is important to choose an appropriate CFI mode to evaluate different circulations with different flow rates (Table 2). Fetal exposure to CFI should be limited to as low as reasonably achievable (ALARA), and the region of interest (ROI) should be kept as small as possible [1].

Using CDFI or HDFI with three-dimensional volume or spatio-temporal image correlation (STIC) in the glass-body mode allows displaying both gray-scale and color information of the heart cycle-related flow events and vessel spatial relationship [4,9]. While the use of the glass-body mode in the evaluation of heart defects has been well described, there is a lack of reporting on its use in the evaluation of non-cardiac abnormalities. Recently, Leung KY has reported a novel application of STIC in the visualization of abdominal precordial veins and intraplacental vascularization in singleton pregnancies [10,11]. The aim of this present review is to discuss the use of color Doppler with three- and four-dimensional (3D and 4D) ultrasonography in the evaluation of extracardiac, placental, umbilical cord and twin abnormalities after a literature review. Selected ultrasound images were used to illustrate the use of the glass-body mode in the assessment of different abnormalities. All 3D ultrasound examinations or STIC acquisitions were performed by the author using a Voluson E10 machine (GE Medical Systems, Zipf, Austria) equipped with a RAB6-D 2–8 MHz volumetric abdominal transducer.

### Methods

A bibliographic search was performed in November 2022 in the Medline/PubMed virtual library using the following descriptors: three-dimensional, Doppler ultrasound, four-dimensional ultrasound, spatiotemporal image correlation technology, and pregnancy. Articles in English, published between 2008 and 2022, available in full text in which the methodologies were case reports, observational studies or literature reviews were reviewed. There were no randomized clinical trials, meta-analyzes or systematic reviews on the use of the glass-body mode in pregnancy.

## 2. Extracardiac Abnormalities

### 2.1. Outflow Tracts

While CDFI with STIC in the glass-body mode can evaluate abnormal anatomy of the fetal heart defects [4], the glass-body mode in HDFI can reveal the spatial relationships of the aortic and pulmonary outflow tracts and arches [12] and facilitate counseling to parents [13]. CDFI/HDFI with 3D ultrasonography can show the ascending aorta, pulmonary trunk, pulmonary veins, superior vena cava, and inferior vena cava, and their relations with the heart. In a fetus with right aortic arch, the trachea is surrounded, and possibly compressed by the main pulmonary trunk, aortic arch, and ligamentum arteriosus. The glass-body mode can demonstrate the course of the trachea and the surrounding vascular structures in an affected fetus (Figure 1).

### 2.2. Brain

The glass-body mode with HDFI can display the origin, course and branching pattern of the normal variants of pericallosal artery with the callosomarginal artery [14] (Figure 2) after a volume acquisition along the sagittal plane. Such display can thus facilitate the prenatal diagnosis of corpus callosum anomalies. In fetal growth restriction (FGR), callosal development is usually altered. Small corpus collosa as well as brain sparing have been independently associated with adverse neonatal and long-term neurobehavioral outcomes. Fetal neurosonography can be used to monitor brain development in fetuses at risk [15,16].

A 3D volume acquisition along the transvers plane allows the reconstruction of the middle cerebral artery and the circle of Willis. 3D power Doppler measurement of cerebral blood flow is significantly altered in growth restricted fetuses [17]. Flow index was higher in the latter than normal growth fetuses. Further studies are required to validate the usefulness of flow index to differentiate FGR from constitutional small fetuses.

Vein of Galen aneurysmal malformation is a rare and complex anomaly. Use of 3D power Doppler rendering can well demonstrate the malformation, its feeders and dilatation of straight sinus [18]. However, 3D power Doppler is not reliable for volume calculation because of the overrepresentation of the aneurysm and vessels, which are hollow structures [18].

With transvaginal approach, 3D PDI/ HDFI can demonstrate small medullary vessels in normal and abnormal fetuses including malformations, hypoxia or infection [19]. These findings may help predict neurological prognosis.

### 2.3. Pulmonary Vessels

The use of STIC combined with color Doppler may help evaluate the involvement of the vascular system in lung dysplasia, which is associated with neonatal mortality [20]. In a fetus with congenital diaphragmatic hernia, this mode can show the location of left hepatic vein [20] which is higher than normal (Figure 3A,B) and hence confirm the presence of ‘liver up’, which is associated with adverse outcomes. Besides, STIC mode can be used to show the proximity of esophageal pouch to the fetal heart in a fetus with esophageal atresia [20].

Congenital diaphragmatic hernia is associated with pulmonary hypoplasia. Various fetal ultrasound parameters have been used to predict neonatal outcome in isolated congenital diaphragmatic hernia. It seems that measuring contralateral pulmonary vascularization by 2D color Doppler or 3D power Doppler is the most accurate ultrasound parameter to predict neonatal death and severe pulmonary arterial hypertension [21]. After fetoscopic tracheal occlusion, the assessment of the increase in lung tissue perfusion by power Doppler as well as the relative increase of lung head ratio seems to improve the prediction of fetal survival [22].

### 2.4. Abdomen

When a targeted examination of the fetal umbilical-portal venous system is required [23], a systematic assessment using CDFI or HDFI in two transverse planes and one sagittal plane, as suggested by Yagel [24], can facilitate the prenatal diagnosis of venous abnormalities, but the assessment remains difficult. Leung KY recently described a new approach of using STIC rendered volume, acquired in the sagittal ductus venosus plane, in glass-body mode to facilitate the assessment of the connection of the hepatic veins, ductus venosus, and inferior vena cava to the fetal heart and their blood flow in a cardiac cycle [10] (Figure 3A and Figure 4A). Tomographic ultrasound imaging (TUI) can be used to study two different venous returns to the right atrium, namely, (a) a right-sided inferior vena cava with flow upwards, and (b) a left-sided ductus venous and the left hepatic vein with flow downwards [10]. The presence of ductus venosus and intrahepatic venous system, a favorable sign, can be demonstrated after acquiring a 3D volume in sagittal view [10]. Three-dimensional/STIC rendered volume can be acquired in the transverse abdominal plane to examine the course of umbilical vein in a normal fetus (Figure 4B) or persistent right umbilical vein (Figure 4C).

2D/3D ultrasonography with power Doppler can demonstrate renal artery. One and both renal arteries are not identified in fetuses with unilateral and bilateral renal agenesis, respectively. Variations of the origin of renal arteries are commonly found in normal pregnancies. However, renal malformations including horseshoe kidney and pelvic kidney can be associated with such vascular variations [25].

### 2.5. Others

Sacrococcygeal teratomas (SCT) are usually highly vascularized and thus can cause high-output cardiac failure, and even fetal death. Evaluation of the severity of SCT can guide appropriate management. In addition to SCT ratio measured by 2D ultrasonography, 3D power Doppler vascularization index % may be a promising prognostic indicator [26].

Prediction of preterm delivery remains difficult. It is well known that a short cervical length measured by 2D ultrasonography is associated with preterm delivery. Addition of 3D transvaginal power Doppler ultrasound may improve the prediction [27]. There are differences in cervical vascularization indices between pregnancies with an asymptomatic short cervix and pregnancies with threatened preterm labor [27].

## 3. Placenta

Placenta accreta spectrum disorder (PAS) is a potentially life-threatening condition associated with severe postpartum hemorrhage. 2D ultrasonography with the use of color Doppler is a major modality to diagnose PAS and assess its severity. Addition of 3D Doppler ultrasound may help detect PAS and predict those PAS requiring hysterectomy [28]. The largest area of confluent 3D power Doppler signal at the uteroplacental interface may be an objective parameter [28].

CFI is helpful in assessing the vascularization in various conditions of placental lakes or cysts which can be benign [29] or are associated with fetal growth restriction [30], features of placental mesenchymal disease [31] or placenta accreta spectrum disorders (PASs) [32]. Although MVFI can allow characterization of the placental microvascular pattern in normal and abnormal pregnancy [33,34,35,36], the information on the direction of blood flow and 3D/STIC are generally not available. Three-dimensional ultrasound with crystal Vue™ rendering can be used to show the details of the vascular branching of a placenta chorangioma [37].

Recently Leung KY described a new use of STIC volume acquisition in HDFI, displayed in the glass-body mode, to show blood flow in the intraplacental branches of the umbilical artery (IPB) and the spiral artery jets in a cardiac cycle [11] (Figure 5A). Compared to normal pregnancies, the IPB were infrequent in a pregnancy at risk of fetal growth restriction, and were long but slender in a pregnancy at a high risk of pre-eclampsia [11] (Figure 5B,C). The length and width of the intraplacental vessels can be measured. Further studies are required to assess their accuracy and predictive values. Besides, the glass-body mode can display the spatial relationship between the placental lakes and vascularization. It seems that the glass-body mode is a promising technique, and further studies are required to investigate its use in the evaluation of intraplacental vascularization.

Pre-eclampsia is a major cause of maternal and fetal morbidity and mortality. 3D power Doppler whole placental volume scanning at 11–14 weeks’ gestation was studied to predict pre-eclampsia [38]. The placental vascularization indices were lower in in pregnant women who would develop pre-eclampsia than those who would not [38]. A recent study showed that both 3D placental volume and 3D power Doppler of placental vascular indices in the first trimester were lower in the women with pre-eclampsia than those women without [39]. In contrast, women with pre-eclampsia had a higher uterine artery pulsatility index than those without [39]. The placental vascular indices were more sensitive, while the placental volume and the uterine artery Doppler were more specific for the prediction of pre-eclampsia [39]. It seems that first-trimester assessment using a combination of these 2D and 3D ultrasound parameters can increase the accuracy of early prediction of pre-eclampsia. Further research is required to evaluate the performance of the various screening models using maternal characteristics, 2D/3D ultrasound, and biochemical markers.

Fetal growth restriction is common and is associated with adverse perinatal outcomes including mortality. It remains difficult to differentiate a growth restricted fetus from a small but normal fetus. The use of 3D power Doppler placental vascular flow indices in the prediction or identification of fetal growth restriction was reviewed [40]. Detection rate was demonstrated only in the third trimester, but prediction rate was not demonstrated in the first trimester [40]. It seems that the use of the Doppler placental vascular flow indices is limited in the prediction of FGR.

## 4. Umbilical Cord

Using CDFI or HDFI can facilitate the detection of a velamentous, marginal or furcate insertion of the umbilical cord into the placenta or vasa previa, which are associated with adverse pregnancy outcomes [37] (Figure 6A). The glass-body mode shows the spatial relationship between a placental surface cyst and marginal cord insertion (Figure 6B). Tomographic ultrasound imaging (TUI) can facilitate identification of the exact site of a velamentous cord insertion (Figure 7). The glass-body mode can be used to show single umbilical artery, cord round neck (Figure 8A,B), umbilical cord knot, abnormal thickening of an umbilical cord with disorganized cord vessels, and umbilical cord vessel aneurysm [37]. STIC can allow visualization of blood flow through umbilical cord vessels in a cine loop.

## 5. Twins

With the advance in ultrasound technology, it is feasible to use 2D and 3D ultrasonography with HDFI to detect placental anastomoses in a monochorionic twin pregnancy, which are the underlying cause of the development of twin–twin transfusion syndrome (TTTS) [41]. The glass-body mode in HDFI and TUI allow visualization of deep arterio-venous anastomoses with a non-accompanying artery and vein branches from two twins entering the same placental lobule (Figure 9A,B). Besides, STIC mode can show blood flow in the anastomoses. Development of an accurate ultrasound-based mapping of placental anastomoses will facilitate the future application of a non-invasive treatment for TTTS [41].

Similar to FGR in singleton pregnancies, the glass-body mode can be used to show differences in intraplacental vascularization between a normal twin and a FGR twin in a monochorionic twin pregnancy (Figure 10A,B). The intraplacental branches are infrequent in the FGR twin.

The glass-body mode in HDFI can facilitate the detection of reverse arterial perfusion through the umbilical cord of the recipient twin in a case of Twin reverse arterial perfusion (TRAP) [42]. Besides, umbilical cord entanglement in a case of monoamniotic twin pregnancy can be shown using this mode.

## 6. Pitfalls of Glass-Body Mode

In general, the glass-body mode, a CFI, can be adversely affected by partial volume effect, limited temporal and velocity resolution, inappropriate angle of insonation, and aliasing [43]. These adverse effects can be ameliorated by appropriate machine settings, proper examination set up, and image optimization. Compared to the glass-body mode with 3D volume acquisition, STIC is limited by small angle of volume acquisition, and is prone to motion artifact because of a longer acquisition time. Apart from the glass-body mode, other advanced 3D technologies including HDlive flow imaging and Crystal Vue can also enhance image quality and facilitate prenatal diagnosis and management of vascular abnormalities [41]. The use of 3D histogram vascularity indexing is not preferred because of its low intra- and interobserver reliability of placental vascularization [44]. The usefulness of linear measurement of intraplacental vascular dimensions requires further investigation.

A recent review highlighted the challenges of 3D/4D CFI [45]. The results of CFI are operator dependent and machine dependent, and hence are of low reproducibility [45]. Proper training on 2D/3D/4D color flow imaging and using a standardized 3D/4D ultrasound protocol are required to improve the reproducibility [45]. There is lack of studies on the learning curve of 3D/4D PD. Whether artificial intelligence can be incorporated into 3D/4D volume acquisition or post processing needs further studies.

## 7. Practical Tips and Application

Image quality in the glass-body mode can be improved by the followed steps. First, a good-quality image of the target structure, for example, the umbilical cord insertion site into the placenta, should be visualized by 2D ultrasonography. Before using color Doppler, it is useful to reduce beam width, depth of penetration and the number of focal points, and hence increase the frame rate and contrast resolution. Appropriate color flow imaging mode should be selected. In general, CDFI is chosen to assess high flow, and HDFI to assess medium or low flow [8].

The angle of insonation is important. To acquire a 3D or STIC volume, an appropriate plane should be chosen. For examples, a cross-sectional plane close to the center of the placenta is used to visualize intraplacental vessels while the sagittal approach is used for visualization of the precordial venous system including ductus venosus. While the region of interest (ROI) should be large enough to include all the targeted vessels, the ROI and the acquisition angle should be kept as narrow as possible to maximize the frame rate during a volume acquisition. Selecting an acquisition time is a trade-off between enhancing the spatial resolution and reducing motion artifacts. The latter can be reduced by acquiring a 3D or STIC volume while the mother is holding her breath and there is no fetal or placental movement. Typical technical settings for a volume acquisition can be set to improve reproducibility.

The acquired 3D or STIC volume is then stored in the ultrasound machine to enable subsequent processing. In the glass-body mode, the rendered image can be displayed in the glass-body fetal heart or the high-definition mode. While 3D volume allows display of static images, STIC volume allows assessment of the targeted vessels during a ‘cardiac cycle’ using a cine loop. The pulse rate of the vessels can be subjectively assessed. During the assessment, it is important not to rotate along the z-axis of a 3D or STIC volume so that the displayed red-blue color signals correspond to the direction of blood flow of the targeted vessels. To locate a pathological condition such as an arterio-venous anastomosis in a twin–twin transfusion syndrome, use of TUI can facilitate a systematic assessment. While adequate diagnostic ultrasound information is to be obtained, the ALARA principle (that is, as low as reasonably achievable) should be observed during a 3D or STIC volume acquisition on a developing fetus especially in the first trimester.

The glass-body mode demonstrates vessel spatial course and its relationship with adjacent non-vascular structures by displaying both gray-scale and color information. The gray-scale information is important for the interpretation of the displayed image, especially in cases of fetal abnormalities. This mode can be used to examine the whole vascular system of the fetus, the intraplacental vessels and the umbilical cord vessels. Both the placenta and the umbilical cord are usually not affected by fetal movements, and thus are less prone to motion artifacts.

Abnormalities of the fetal vascular system are often complex, requiring gray-scale imaging and color or power Doppler ultrasound to assess the abnormalities. 2D, 3D or 4D ultrasonography are non-invasive without radiation or contrast media. Although 2D ultrasound can allow examination of fetal vascular abnormalities, there are limitations in the examination when the examined vessels do not follow a straight course or lie in a 2D plane. In that case, the examiner needs to reconstruct mentally a 3D image of the examined vessels. On the other hand, 3D ultrasound rendered mode can allow display of complex vascular structures, whereas TUI can facilitate a systematic examination along multiple parallel sagittal, transverse or coronal planes.

## 8. Conclusions

Displaying the glass-body mode after 3D or STIC volume acquisition in CDFI or HDFI can facilitate evaluation of various fetal non-cardiac, placental, umbilical cord and twin abnormalities, in addition to cardiac abnormalities (Table 3). In particular, the glass-body mode can be used to demonstrate the normal and abnormal course of (a) the vascular structures surrounding trachea, (b) pericallosal artery with the callosomarginal artery, (c) pulmonary vessels, and (d) umbilical-portal venous system. While 3D volume acquisition can display static images, the use of STIC technique can demonstrate heart cycle-related flow events. The use of TUI can facilitate a systematic assessment and localization of a vascular pathology. The glass-body mode is complementary to conventional 2D ultrasonography. Further studies are required to investigate use of the glass-body mode in the assessment of intraplacental vascularization in singleton and twin pregnancies.

## Figures and Tables

**Figure 1 healthcare-11-00488-f001:**
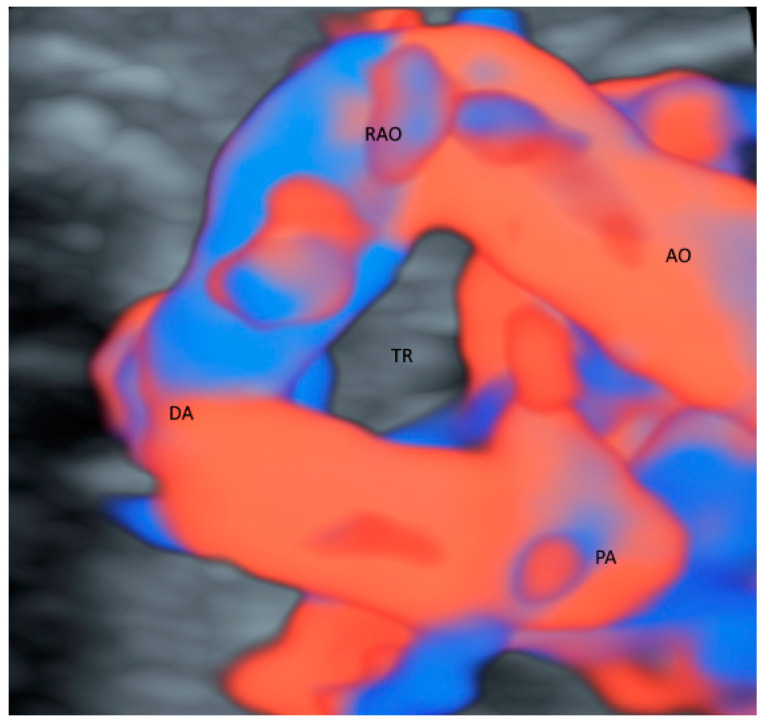
Glass-body rendering mode with high-definition Doppler showing a fetus with right aortic arch at 36 weeks’ gestation. Noted that the trachea (TR) is surrounded by Aorta (AO), right aortic arch (RAO), pulmonary artery (PA) and ductus arteriosus (DA).

**Figure 2 healthcare-11-00488-f002:**
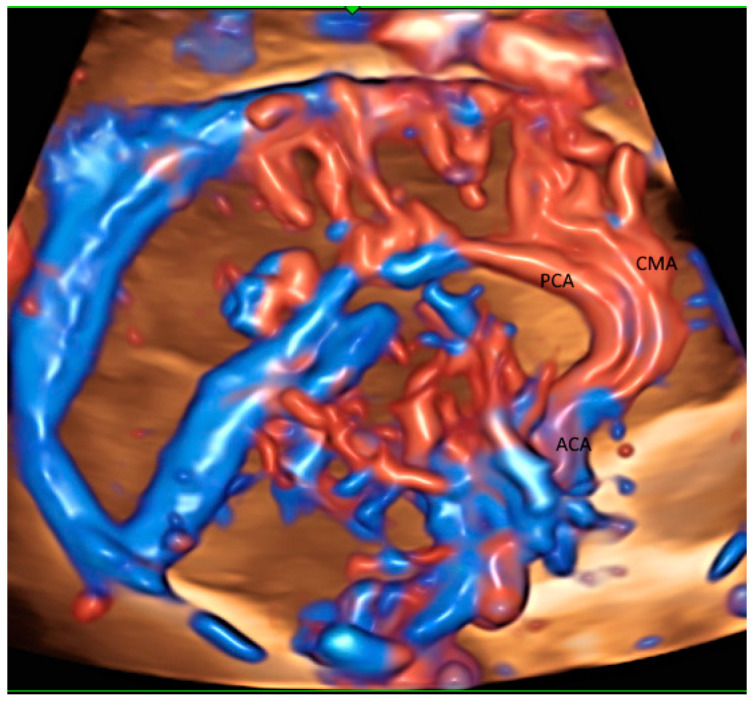
Glass-body rendering mode with high-definition Doppler showing the normal course of pericallosal artery (PCA) and its main branch, and callosomarginal artery (CMA) in a fetus with severe fetal growth restriction at 31 weeks’ gestation. ACA, anterior cerebral artery.

**Figure 3 healthcare-11-00488-f003:**
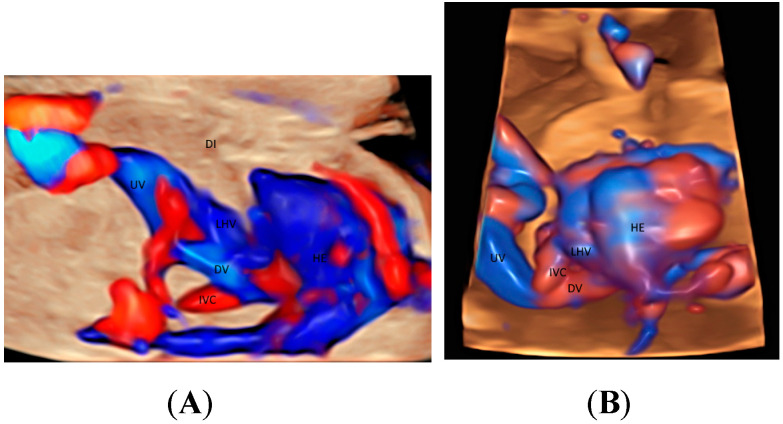
Glass-body rendering mode with high-definition Doppler showing a normal fetus (**A**), and a fetus with left diaphragmatic hernia with ‘liver up’ (**B**), both at 21 weeks’ gestation. Note the difference in the position of the left hepatic vein (LHV) draining into the right atrium (RA) between (**A**,**B**). DI, diaphragm; Ductus venosus (DV); HE, heart; IVC, inferior vena cava; UV, umbilical vein.

**Figure 4 healthcare-11-00488-f004:**
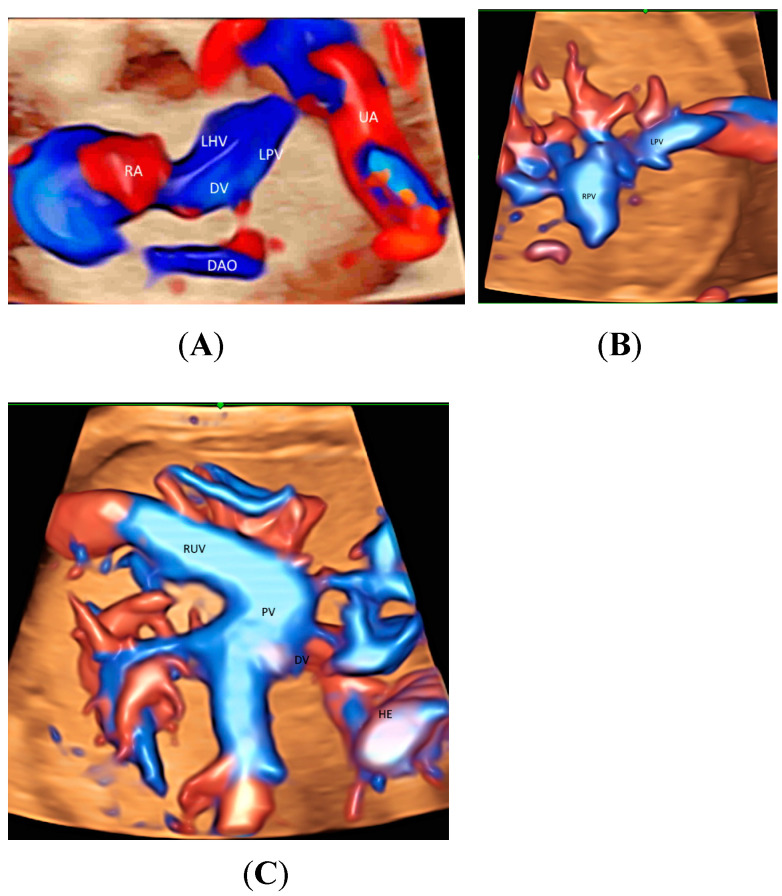
Spatiotemporal image correlation volume acquisition in color Doppler displayed in glass-body mode in a longitudinal view of a normal fetus at 14 weeks’ gestation (**A**), in a transverse view of a normal fetus at 21 weeks (**B**), and of a fetus with persistent right umbilical vein at 29 weeks (**C**). DAO, descending aorta; DV, ductus venosus; HE, heart; LHV, left hepatic vein; LPV, left portal vein; PV, portal vein; RA, right atrium; RPV, right portal vein; RUV, right umbilical vein; UA, umbilical artery.

**Figure 5 healthcare-11-00488-f005:**
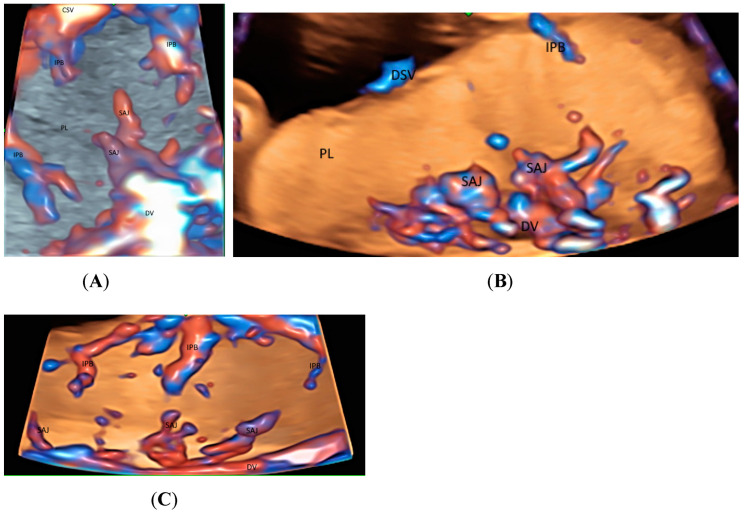
Three-dimensional or spatiotemporal image correlation volume acquisition in high-definition flow, displayed in glass-bode mode in a normal pregnancy (**A**), in a pregnancy at an increased risk of fetal growth restriction (**B**), and in a pregnancy at a high risk of pre-eclampsia (**C**), all at 20 weeks’ gestation. In (**A**), note the spiral artery jets (SAJ) opening into the intervillous space of the placenta (PL), and the intraplacental branches (IPB) of the umbilical artery with primary and secondary order branches. In (**B**), only one small primary order branch is noted without any secondary order branches. In (**C**), IPB are long but slender with few secondary or tertiary order branches. CSV, chorionic surface vessels; DV, decidual vessels; PL, placenta.

**Figure 6 healthcare-11-00488-f006:**
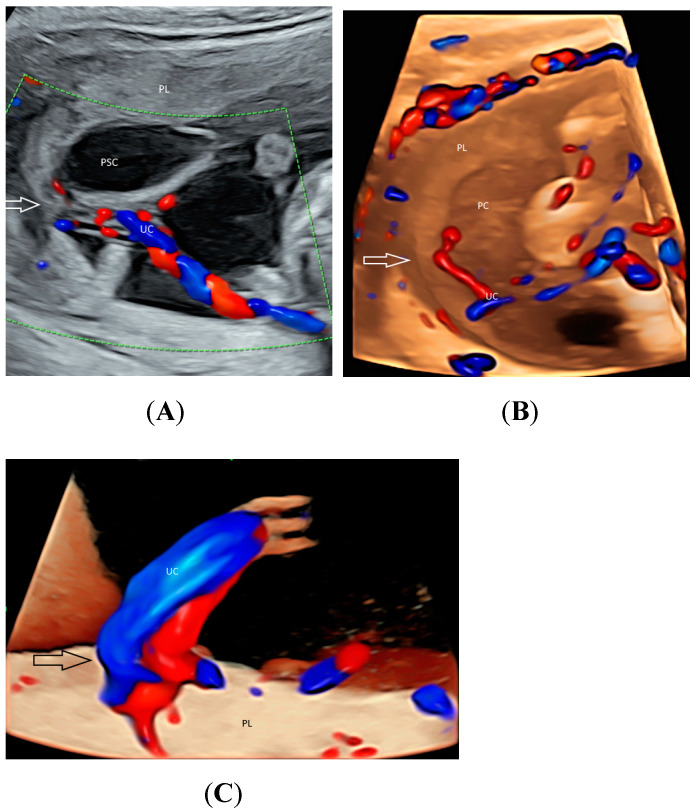
Two-dimensional color-flow imaging (**A**) and glass-body rendering mode with color-flow Doppler (**B**) showing a placental surface cyst (PSC) associated with marginal insertion (arrow) of the umbilical cord (UC) into the placenta (PL) at 17 weeks’ gestation. For comparison, a central cord insertion (arrow) is noted in another fetus (**C**) at 21 weeks’ gestation.

**Figure 7 healthcare-11-00488-f007:**
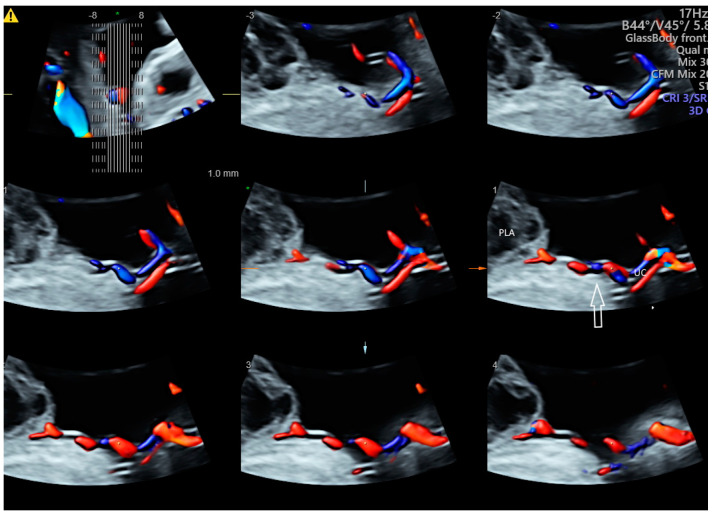
Tomographic ultrasound imaging of three-dimensional volume of the umbilical cord (UC) in color Doppler demonstrating velamentous cord insertion into a site distance from the placenta edge (arrow). PLA, placental lake.

**Figure 8 healthcare-11-00488-f008:**
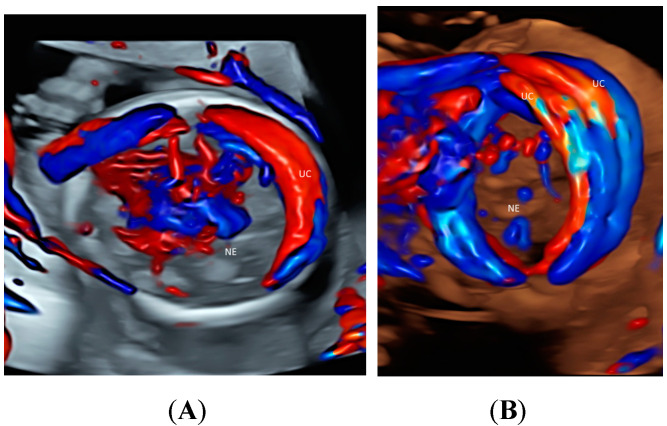
Glass-body rendering mode with color Doppler showing nuchal cord with umbilical cord (UC) wrapping around the neck (NE) once in a fetus (**A**) and twice in a fetus (**B**), both at 20 weeks’ gestation.

**Figure 9 healthcare-11-00488-f009:**
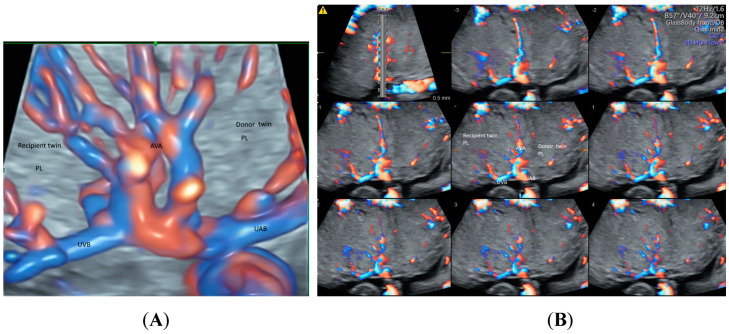
Demonstration of deep arteriovenous anastomosis in a complicated monochorionic twin pregnancy at 25 weeks’ gestation by spatiotemporal image correlation volume acquisition in high-definition Doppler displayed in glass-body mode (**A**) and tomographic ultrasound imaging (**B**). Note umbilical artery (UAB) and vein branches (UVB) from two twins converged into the same placental lobule (PL), and the deep arteriovenous anastomoses (AVA).

**Figure 10 healthcare-11-00488-f010:**
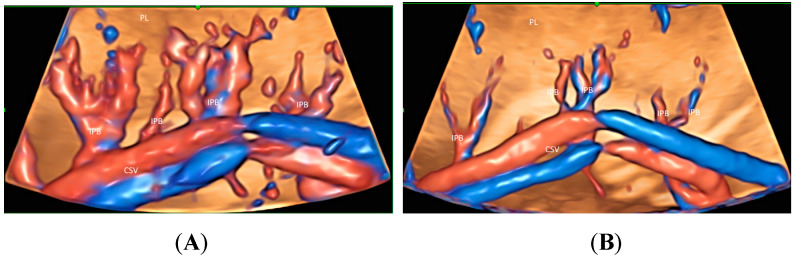
Three-dimensional volume acquisition in high-definition flow, displayed in glass-bode mode in a monochorionic twin pregnancy with selective growth restriction (FGR) at 27 weeks’ gestation. Note the difference in the intraplacental branches (IPB) of the umbilical artery between the twin fetus with normal growth (**A**) and that with FGR (**B**). Branches are infrequent in (**B**).

**Table 1 healthcare-11-00488-t001:** Characteristics of different modes of color flow imaging.

	CDFI	PDI	HDFI	MVFI
Sensitive to/ Use with	Rapid flow or large vessels	Medium flow or medium-size vessels	Slow to medium flow or small to medium sized vessels	Slow flow or small vessels
Insonation angle	Dependent	Independent	Dependent	Independent
Aliasing	Present	Absent	Present	Absent
Flash artifacts	Present	Present	Present	Present
Motion artifacts	Present	Present	Present	Reduced
Directional information	Bi-directional	Uni-directional	Bi-directional	mostly uni-directional
Spatial resolution	Low	Low	High	High
Radiant flow	Applicable	Applicable	Applicable	Applicable
3D/4D with STIC, glass-body mode	Applicable	Applicable	Applicable	Mostly not applicable

Color Doppler flow imaging (CDFI), power Doppler imaging (PDI), high-definition flow imaging (HDFI), microvascular flow imaging (MVFI), Spatio-temporal Image Correlation (STIC).

**Table 2 healthcare-11-00488-t002:** Various CFI modes in the assessment of fetal and placental circulations.

Fetus/Placenta	CDFI	PDI/HDFI	MVFI
Heart	Across atrioventricular and semilunar valves	Pulmonary veins and other small/low-flow vessels	First trimester
Brain	Circle of Willis	Pericallosal artery Circle of Willis	Medullary veins and other small vessels
Abdomen and other parts	Precordial veins	Umbilical arteries, Renal arteries Splenic artery Hepatic arteries	Small vessels in lung, liver, spleen, kidney, adrenal gland and limbs
Umbilical cord	Cord vessels and insertion site	Cord vessels and insertion site	
Placenta	Placenta accreta spectrum disorders	Chorionic and decidual vessels, Spiral artery jets Placental vascular anastomoses in twin pregnancies	Stem villous vessels

Color Doppler flow imaging (CDFI), power Doppler imaging (PDI), high-definition flow imaging (HDFI), microvascular flow imaging (MVFI).

**Table 3 healthcare-11-00488-t003:** Use of color Doppler flow imaging (CDFI)/ high-definition flow imaging (HDFI) with 3-dimensional ultrasonography (3DUS)/ spatiotemporal image correlation (STIC) in the prenatal evaluation of fetal cardiac and placental abnormalities.

Fetus/Placenta	CDFI/HDFI	3DUS/ STIC	Examples of Abormalities
Heart/chest	High flow (CDFI) Small/low flow (HDFI)	GBM	Abnormal outflow tracts Course of trachea in right aortic arch Pulmonary vessels in lung dysplasia Liver vessels in diaphragmatic hernia Esophageal pouch in esophageal atresia
Brain	Pericallosal artery (HDFI)	GBM	Altered callosum development in FGR
Abdomen	Precordial veins (HDFI)	GBM TUI	Anomalies of the precoridal venous system Persistent right umbilical vein Abnormal venous returns to the right atrium
Umbilical cord	Cord vessels and insertion site (HDFI)	GBM TUI	Velamentous or marginal cord insertion, vasa previa, cord round neck, single umbilical artery, umbilical cord knot, abnormal thickening
Placenta	High flow (CDFI) Chorionic and decidual vessels, Spiral artery jets (HDFI)	GBM	Different intraplacental vascularization in PAS, FGR, and PET
Twin	Placental vascular anastomoses (HDFI) Intraplacental branches of umbilical artery (HDFI)	GBM TUI	Placental anastomoses in TTTS Different intraplacental vascularization in selective FGR TRAP, Cord entanglement in monoamniotic twin

Fetal growth restriction (FGR), Glass-body mode (GBM), placental accreta spectrum disorders (PAS), pre-eclampsia (PET), tomographic ultrasound imaging (TUI), twin reversed arterial perfusion (TRAP), twin-twin transfusion syndrome (TTTS).

## Data Availability

Data is contained within the article.

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
