# Peer review of "Application of Color Doppler with 3- and 4-Dimensional Ultrasonography in the Prenatal Evaluation of Fetal Extracardiac and Placental Abnormalities"

_healthcare, 2023, doi:10.3390/healthcare11040488_

Round 1

Reviewer 1 Report

The text requests major changes.

It is not clear which type o paper is this: review or retrospective?

In the text are presented a lot of images from the personal casuitry. This approach is concordant with a retrospective study, but is not allowed in a review paper.

If the author will decide to keep the review format, I suggest a supplementation of bibliographic sources.

If the author will change the format to retrospective paper, the text request major changes, with application of standard type: introduction, methodes, cohort of patients, results, discussion, conclusion.

In the text analysed I found some mistakes and I suggest to read the following list:

row 8 - are used two abbreviations without explanations; I suggest to write first what meaning the abbreviations

row 23 - change [R8] with [8]

row 34 - the sentence part: ” its use in the evaluation of non-cardiac abnormalities is not” is incomplete

in table 1 in the row concerning ”direction information” I suggest to use corresponding adjectives: „directional” instead of ”direction” „bidirectional” instead of ”bi-direction” „unidirectional” instead of ”uni-direction”

row 51 - the sentence part: ” and facilitate parental counseling to parents” is not clear

row 60-62 - the sentence: ” The glass-body mode with HDFI can display the origin, course and branching pattern of the normal variants of pericallosal artery with the callosomarginal artery [14] (Figure 2), can thus facilitate the prenatal diagnosis of corpus callosum anomalies.” request a splitting in two sentences

row 62 - 63 - the sentence: ” In fetal growth restriction (FGR), callosal development was altered.” I suggest to replace ”... callosal development was altered” with ” callosal development is usually altered”

row 115 - the word ”noel” is incorrect used; probably the correct word is ”new”

row 120 - the word ” Glass-body” requests the replacement of caps lock

row 140 - replace ” can used” with ” can be used”

Author Response

Thanks for your comments. They are useful to improve the quality of my manuscript. My response is as follow: 

The text requests major changes.

  1. It is not clear which type o paper is this: review or retrospective?

In the text are presented a lot of images from the personal casuitry. This approach is concordant with a retrospective study, but is not allowed in a review paper.

If the author will decide to keep the review format, I suggest a supplementation of bibliographic sources.

If the author will change the format to retrospective paper, the text request major changes, with application of standard type: introduction, methodes, cohort of patients, results, discussion, conclusion.

Response: I prefer to keep the review format, and have revised the text as: ‘The aim of this present review is to discuss the use of the glass-body mode in the evaluation of non-cardiac, placental, umbilical cord and twin abnormalities after a literature review. Selected ultrasound images were used to illustrate the use of the glass-body mode in the assessment of different abnormalities. All three-dimensional ultrasound examinations or STIC acquisitions were performed by the author using a Voluson E10 machine (GE Medical Systems, Zipf, Austria) equipped with a RAB6-D 2–8 MHz volumetric abdominal transducer.

  1. In the text analysed I found some mistakes and I suggest to read the following list:

row 8 - are used two abbreviations without explanations; I suggest to write first what meaning the abbreviations

Response: I have revised the text as: ‘Using color Doppler flow imaging or high-definition flow imaging with three-dimensional volume or spatio-temporal image correlation (STIC) in the glass-body mode allows displaying both gray-scale and color information of the heart cycle-related flow events and vessel spatial relationship.’

  1. row 23 - change [R8] with [8]

Response: Revised accordingly

  1. row 34 - the sentence part: ” its use in the evaluation of non-cardiac abnormalities is not” is incomplete

Response: I have revised the text as: ‘While the use of the glass-body mode in the evaluation of heart defects has been well described, there is a lack of reporting on its use in the evaluation of non-cardiac abnormalities.’

  1. in table 1 in the row concerning ”direction information” I suggest to use corresponding adjectives: „directional” instead of ”direction” „bidirectional” instead of ”bi-direction” „unidirectional” instead of ”uni-direction”

Response: revised accordingly.

  1. row 51 - the sentence part: ” and facilitate parental counseling to parents” is not clear

Response: I have revised the text as: ‘…facilitate counseling to parents.’

  1. row 60-62 - the sentence: ” The glass-body mode with HDFI can display the origin, course and branching pattern of the normal variants of pericallosal artery with the callosomarginal artery [14] (Figure 2), can thus facilitate the prenatal diagnosis of corpus callosum anomalies.” request a splitting in two sentences

Response: I have revised the second part of the sentence as: ‘Such display can thus facilitate the prenatal diagnosis of corpus callosum anomalies.’ 

  1. row 62 - 63 - the sentence: ” In fetal growth restriction (FGR), callosal development was altered.” I suggest to replace ”... callosal development was altered” with ” callosal development is usually altered”

Response: revised accordingly.

  1. row 115 - the word ”noel” is incorrect used; probably the correct word is ”new”

Response: revised accordingly.

  1. row 120 - the word ” Glass-body” requests the replacement of caps lock

Response: revised accordingly.

  1. row 140 - replace ” can used” with ” can be used”

Response: revised accordingly.

Reviewer 2 Report

I read the article sent to me for review with great interest. I fully agree with the authors that the display of the body glass mode after 3D or STIC volume acquisition in CDFI or HDFI facilitate ultrasound diagnostics of pregnancy and may be a useful supplement to conventional 2D ultrasound.

The manuscript has an educational value for physicians trained in ultrasonography. Therefore, my opinion - obviously subjective - was very positive. In my opinion, the educational value of the manuscript can be taken as the strength of this article. The value of the manuscript is enriched by numerous ultrasound images of quite good quality. On the other hand, there are of course the weaknesses of this paper.  It is necessary to mention the lack of a chapter describing the methodology, on the basis of which criteria, the articles on the basis of which this review was carried out were selected- as the authors emphasize, this is a review article and when you read the manuscript, you don't get that impression. I can suggest the authors to include such a section in the manuscript. The section entitled "Conclusion" is too laconic - it is worth emphasizing the practical aspects of using the glass-body mode in the assessment of fetal anatomy. The overall impression that the reviewed paper made on me is positive.

Author Response

Thanks for your comments. They are useful to improve the quality of my manuscript. My response is as follow: 

I read the article sent to me for review with great interest. I fully agree with the authors that the display of the body glass mode after 3D or STIC volume acquisition in CDFI or HDFI facilitate ultrasound diagnostics of pregnancy and may be a useful supplement to conventional 2D ultrasound.

The manuscript has an educational value for physicians trained in ultrasonography. Therefore, my opinion - obviously subjective - was very positive. In my opinion, the educational value of the manuscript can be taken as the strength of this article. The value of the manuscript is enriched by numerous ultrasound images of quite good quality. On the other hand, there are of course the weaknesses of this paper. 

  1. is necessary to mention the lack of a chapter describing the methodology, on the basis of which criteria, the articles on the basis of which this review was carried out were selected- as the authors emphasize, this is a review article and when you read the manuscript, you don't get that impression. I can suggest the authors to include such a section in the manuscript.

Response: I have added the following text: ‘

Methods

A bibliographic search was performed in November 2022 in the Medline/PubMed virtual library using the following descriptors: three-dimensional, Doppler ultrasound, four-dimensional ultrasound, spatiotemporal image correlation technology, and pregnancy. Articles in English, published between 2008 and 2022, available in full text in which the methodologies were case report, observational studies or literature reviews were reviewed. There were no randomized clinical trials, meta-analyzes or systematic reviews on the use of the glass-body mode in pregnancy.’

  1. The section entitled "Conclusion" is too laconic - it is worth emphasizing the practical aspects of using the glass-body mode in the assessment of fetal anatomy. The overall impression that the reviewed paper made on me is positive.

Response: In the Conclusion, I have added the following text: ‘The glass-body mode can be used to demonstrate the normal and abnormal course of (a) the vascular structures surrounding trachea, (b) pericallosal artery with the callosomarginal artery, (c) pulmonary vessels, and (d) umbilical-portal venous system. The use of STIC can demonstrate heart cycle-related flow events. The use of TUI can facilitate systematic assessment and localization of a vascular pathology.’

Reviewer 3 Report

In this study Authors perform a review of the possible application of 3D and 4 D color Doppler in the diagnosis of extracardiac anomalies

The subject is of interest and i would like to congratuate with Authors for their effort

My comments are as follows

1)in the title i will add extracardiac fetal anomalies

2) add a table summarizing the anomalies in which there is an advance in using 3D/4D

3)considers in the discussion the paper by Santana 22 Diagnostics

Author Response

Thanks for your comments. They are useful to improve the quality of my manuscript. My response to the comments are as follow:

In this study Authors perform a review of the possible application of 3D and 4 D color Doppler in the diagnosis of extracardiac anomalies

The subject is of interest and i would like to congratuate with Authors for their effort

My comments are as follows

1)in the title i will add extracardiac fetal anomalies

Response: Revised accordingly.

2) add a table summarizing the anomalies in which there is an advance in using 3D/4D

Response: In the Conclusions, I have added Table 3, and revised the text as: ‘HDFI can facilitate evaluation of various fetal non-cardiac, placental, umbilical cord and twin abnormalities (Table 3), in addition to cardiac abnormalities.’

Table 3 Use of color Doppler flow imaging (CDFI)/ high-definition flow imaging (HDFI) with 3-dimensional ultrasonography (3DUS)/ spatiotemporal image correlation (STIC) in the prenatal evaluation of fetal cardiac and placental abnormalities

Fetus/ placenta

CDFI/HDFI

3DUS/

STIC

Examples of abormalities

Heart/chest

High flow (CDFI)

Small/low flow (HDFI)

GBM

Abnormal outflow tracts

Course of trachea in right aortic arch

Pulmonary vessels in lung dysplasias

Liver vessels in diaphragmatic hernia

Esophageal pouch in esophageal atresia

Brain

Pericallosal artery (HDFI)

GBM

Altered callosum development in FGR

Abdomen

Precordial veins (HDFI)

GBM

TUI

Anomalies of the precoridal venous system

Persistent right umbilical vein

Abnormal venous returns to the right atrium

Umbilical cord

Cord vessels and insertion site (HDFI)

GBM

TUI

Velamentous or marginal cord insertion, vasa previa, cord round neck, single umbilical artery, umbilical cord knot, abnormal thickening

Placenta

High flow (CDFI)

Chorionic and decidual vessels, Spiral artery jets (HDFI)

GBM

Different intraplacental vascularization in PAS, FGR, and PET

Twin

Placental vascular anastomoses (HDFI)

Intraplacental branches of umbilical artery (HDFI)

GBM

TUI

Placental anastomoses in TTTS

Different intraplacental vascularization in selective FGR

TRAP, Cord entanglement in monoamniotic twin

Fetal growth restriction (FGR), Glass-body mode (GBM), placental accreta spectrum disorders (PAS), pre-eclampsia (PET), tomographic ultrasound imaging (TUI), twin reversed arterial perfusion (TRAP), twin-twin transfusion syndrome (TTTS)

3)considers in the discussion the paper by Santana 22 Diagnostics

Response: I have added a new reference:

‘Santana, E.F.M.; Castello,R.G.; Rizzo G;, Grisolia,G.; Júnior,E.A.; Werner,H.; Lituania, M.; Tonni, G. Placental and Umbilical Cord Anomalies Diagnosed by Two- and Three-Dimensional Ultrasound. Diagnostics (Basel). 2022;12:2810.’

In the first paragraph of section 3. Placenta, I have added the following text: ‘Three-dimensional ultrasound with crystal Vue™ rendering can be used to show the details of the vascular branching of a placenta chorangioma [28].’

In section 4 Umbilical cord, I have revised the text as:’…CDFI or HDFI can facilitate the detection of a velamentous, marginal or furcate insertion of the umbilical cord into the placenta or vasa previa which are associated with adverse pregnancy outcomes (Figure 6A) [28]…. Glass-body mode can be used to show single umbilical artery, nuchal cords (Figures 8A and 8B), umbilical cord knot, abnormal thickening of an umbilical cord with disorganized cord vessels, and umbilical cord vessel aneurysm [28].’

In the paragraph “Pitfalls of glass-body mode’, I have added the following text: ‘Apart from glass-body mode, other advanced 3D technologies including HDlive flow imaging and Crystal Vue can enhance image quality and facilitate prenatal diagnosis of vascular abnormalities [28].’

Round 2

Reviewer 1 Report

This form of paper is better in comparison with the previously form.